# Inverted-Bearing Reverse Shoulder Arthroplasty: Consequences on Scapular Notching and Clinical Results at Mid-Term Follow-Up

**DOI:** 10.3390/jcm11195796

**Published:** 2022-09-29

**Authors:** Alessandro Castagna, Mario Borroni, Luigi Dubini, Stefano Gumina, Giacomo Delle Rose, Riccardo Ranieri

**Affiliations:** 1Department of Biomedical Sciences, Humanitas University, Via Rita Levi Montalcini 4, Rozzano (Mi), 20090 Milan, Italy; 2IRCCS Humanitas Clinical and Research Center, Via Manzoni 56, Rozzano (Mi), 20089 Milan, Italy; 3Department of Anatomy, Histology, Forensic Medicine and Orthopaedics, Sapienza University of Rome, Piazzale Aldo Moro 5, 00185 Roma, Italy; 4Istituto Clinico Ortopedico Traumatologico (ICOT), Via Franco Faggiana 1668, 04100 Latina, Italy

**Keywords:** reverse shoulder arthroplasty, cuff tear arthropathy, polyethylene, scapular notching, range of motion, larger glenosphere

## Abstract

Background: Scapular notching following reverse shoulder arthroplasty (RSA) is caused by both biological and mechanical mechanisms. Some authors postulated that osteolysis that extends over the inferior screw is caused mainly by biological notching. Inverted-bearing RSA (IB-RSA) is characterized by a polyethylene glenosphere and a metallic humeral liner, decreasing the poly debris formation and potentially reducing high grades of notching. This study aims to report the results of IB-RSA on a consecutive series of patients at mid-term follow-up, focusing on the incidence of Sirveaux grade 3 and 4 scapular notching. Methods: A retrospective study on 78 consecutive patients who underwent primary IB-RSA between 2015–2017 was performed. At a 4 years minimum follow-up, 49 patients were evaluated clinically with Constant score (CS), Subjective shoulder value (SSV), American Shoulder and Elbow score (ASES), pain and range of motion, and with an X-ray assessing baseplate position (high, low), implant loosening, and scapular notching. Results: At a mean follow-up of 5.0 ± 0.9, all the clinical parameters improved (*p* < 0.05). One patient was revised for an infection and was excluded from the evaluation, two patients had an acromial fracture, and one had an axillary neuropraxia. Scapular notching was present in 13 (27%) patients (six grade 1, seven grade 2) and no cases of grade 3 and 4 were observed. Scapular nothing was significantly associated with high glenoid position (*p* < 0.001) and with lower CS (70 ± 15 vs. 58 ± 20; *p* = 0.046), SSV (81 ± 14 vs. 68 ± 20; *p* = 0.027), ASES (86 ± 14 vs. 70 ± 22; *p* = 0.031), and anterior elevation (148 ± 23 vs. 115 ± 37; *p* = 0.006). A 44 mm- compared to 40 mm-glenosphere was associate with better CS (63 ± 17 vs. 78 ± 11; *p* = 0.006), external (23 ± 17 vs. 36 ± 17; *p* = 0.036), and internal rotation (4.8 ± 2.7 vs. 7.8 ± 2.2; *p* = 0.011). Conclusions: IB-RSA is a safe and effective procedure for mid-term follow-up. Inverting biomaterials leads to a distinct kind of notching with mainly mechanical features. Scapular notching is associated with a high baseplate position and has a negative influence on range of motion and clinical outcome.

## 1. Introduction

Scapular notching is a common phenomenon associated with reverse shoulder arthroplasty (RSA) with a variable rate of 4.6–50.8% and up to 96% [1,2]. It can be considered a consequence of the inverted biomechanics of the shoulder, creating a semi-constrained joint. From a pathophysiological point of view, there are two different types of notching [3]: (1) mechanical notching, secondary to the contact of the humeral liner with the scapular pillar during movements in adduction, extension, and external rotation [4]; (2) biological notching, which is a chronic foreign-body reaction caused by polyethylene (PE) debris formation, leading to progressive osteolysis [5]. The radiological Sirveaux classification aims to quantify scapular notching, identifying four grades according to the amount of osteolysis [6] (Figure 1): some authors postulated that grades 1 and 2 are mainly due to mechanical notching, while grades 3 and 4, when it occurs above the inferior screw, are likely the results of the biological reaction [3].

The clinical impact of scapular notching is controversial: some authors have found that notching has no influence on functional score [2,7,8], while other authors showed that it is associated with lower clinical results [6,9,10,11], and recently, Spiry et al. demonstrated a significant relationship between severe notching and late glenoid loosening [12].

For these reasons, since the introduction of the classic Grammont design, different solutions have been developed to avoid this complication and improve clinical results. Firstly, optimal glenoid positioning is a crucial factor to minimize this complication [5,8,11]. Secondly, lateralizing implants on both glenoid and humeral sides have shown to decrease the rate of notching [1,3,13,14,15,16,17].

While all these solutions act mainly on mechanical notching, an alternative solution is the inverted-bearing RSA (IB-RSA), where the prosthesis is characterized by a glenosphere made of PE and a metallic humeral liner [13,18,19]. This solution should theoretically minimize the wear of the PE, which is mainly due to the contact of the PE humeral liner with the scapula in the classic design [18], and decrease the biological component of scapular notching.

The primary endpoint of the study is to report the results of IB-RSA on a consecutive series of patients at mid-term follow-up, focusing on the incidence of grade 3 and 4 scapular notching. Secondary endpoints are other radiological and clinical outcomes. The hypothesis is that IB-RSA is a safe procedure and avoids scapular notching higher than Sirveaux grade 2.

## 2. Materials and Methods

This is a monocentric retrospective study on consecutive patients who underwent IB-RSA between 2015–2017 and evaluated at minimum 4 years follow-up. Patients treated for cuff tear arthropathy, primary osteoarthritis, inflammatory arthritis, and fracture sequelae who were available for clinical and radiological follow-up were included. We excluded patients who were operated on for acute fractures, patients treated for revision arthroplasty, and patients who received associated glenoid bone graft or metal glenosphere. A total of 78 patients who met the inclusion and exclusion criteria were operated on during the index period. Among these, 6 were dead, 17 were lost or impossible to contact, and 6 refused the control, leaving 49 patients reviewed clinically and radiologically at a mean follow-up of 5.0 ± 0.9 years. Thirty-three (67%) were female and 16 (33%) male. The mean age at surgery was 71 ± 7 years. The most common indication was rotator cuff arthropathy (49%), followed by primary osteoarthritis (31%), massive rotator cuff tear (12%), and fracture sequelae (8%).

### 2.1. Surgical Procedure

The SMR metal baseplate has a central peg and two screws, and the SMR long stem is an inlay design with a 150° neck-shaft angle. The SMR Reverse HP has a 40- or 44-mm diameter to improve ROM and it is characterized by an inversion of the materials with the aim to reduce polyethylene debris and, by a smart design (inferior sphere extension and superior narrowing), to facilitate the implantation and improving range of motion. It is made of a highly cross-linked PE (X-UHMWPE) and is coupled with CoCrMo liners. The glenoid implant provides intrinsic lateralization of the center of rotation of 5.2 mm. The glenosphere also presents a 4mm eccentricity option, but it is not utilized in primary cases at our institution. The humeral stem is implanted with 0° of retroversion using a forearm ancillary guide. A 40 mm glenosphere was used in 39 cases and a 44 mm glenopshere in 10 cases.

All the patients received both a general anesthetic and an interscalene block. The operation was performed in beach chair position, through a deltopectoral approach. The SMR RSA (LimaCorporate S.p.A, 33038 Villanova di San Daniele del Friuli, Udine, Italy) with the HP glenosphere was implanted in all the cases (Figure 2).

### 2.2. Clinical and Radiological Evaluation

Clinical evaluation performed pre- and post-operatively included the Constant–Murley score (CS) [20], the Subjective Shoulder Value (SSV) [21], the American Shoulder and Elbow Surgeon (ASES) score [22], the visual analogue scale (VAS) for pain and the range of motion (ROM) in term of active anterior elevation (AE), active external rotation (ER) in position 1, and active internal rotation (IR) (Constant-Murley subcategory). All complications and reoperation were recorded.

At last follow-up, radiographical evaluation was performed on the true anteroposterior projection on the glenohumeral joint line plane, with the humerus in neutral, external, and internal rotation. All the images were evaluated by two senior orthopedic residents trained in shoulder surgery. No attempt was made to determine the reliability of the observations, and when differences in assessments were noted, the observers reached a consensus. The positioning of the glenoid implant and the presence of radiolucent lines (RLL) were evaluated according to the classification system previously described for this baseplate in the anatomic prosthesis [23]. Loosening was considered to be present if the glenoid component had progressively migrated, as demonstrated by shift, tilt, or subsidence, or if complete radiolucency ≥2 mm was present in each zone [24]. On the humeral side, humeral RLL and loosening and partial or total greater tuberosity (GT) resorption were evaluated according to Melis et al. [24]. Inferior scapular notching was graded according to the classification system of Sirveaux et al. [6]. Pillar spurs and ossification, either individually or together, in the scapular-humeral space were recorded. According to the position of the inferior margin of the metallic baseplate in relation to the inferior border of the glenoid, the baseplate was evaluated to be high (inferior margin higher than inferior glenoid border) or low (inferior margin flush or lower than inferior glenoid border). Baseplate inclination was measured as the angle between the baseplate plane (line passing through the inferior e superior margin of the baseplate) and the supraspinatus fossa [16].

### 2.3. Statistical Analysis

The d’Agostino-Pearson test was used to analyze the distribution of the data collected, after which a paired t-test or the Mann-Whitney test was used to evaluate for statistical significance. Qualitative data were compared using the Chi2 and Fisher exact tests. Statistical analysis was performed with EasyMedStat software (Version 3.20; Amiens, France; www.easymedstat.com (accessed on 24 September 2022)).

## 3. Results

### 3.1. Clinical Results

Among the 49 patients, one patient was revised for infection and was excluded in the final evaluation, leaving 48 patients available for the study. Two patients had an acromial fracture and were treated conservatively. One patient suffered a postoperative infection which was revised in two stages. One patient had an axillary neuropraxia, which partially recovered. No loosening and no component disassembly was observed at the last follow-up.

All the clinical scores and range of motion improved at the last follow-up compared to the preoperative status (Table 1).

Patients with 44 mm glenosphere showed a significantly higher CS and range of motion compared to patients with 40 mm glenosphere (Table 2).

### 3.2. Radiological Results

At the radiological evaluation, an RLL <2 mm around the glenoid was observed in five (10%) cases and ≥2 mm in a single zone in one case, which appeared to be progressive. In three cases we observed an initial subsidence of the base plate due to incomplete glenoid preparation, which stabilized within the first year without any progressive change at the last follow-up (Figure 3).

An RLL <2 mm around the humerus was observed in 10 (21%) cases and was confined only to position 4 in 7 of the 10 cases. The distribution of RLL is shown in Figure 4.

GT was partially resorbed in 11 (23%) cases and totally in one (2%) case. Calcar was partially resorbed in 12 (25%) cases and totally in two (4%) cases. Cortical narrowing in zones 2 and 6 was present in 38 (79%) cases with 13 (27%) patients showing spot welds or condensation lines around the stem tip.

### 3.3. Scapular Notching

Scapular notching was present in 13 (27%) patients: 6 cases were grade 1 and 7 cases were grade 2. No cases of grades 3 and 4 were observed. All cases presented a bone spur formation at the scapular neck. Notching was significantly associated with high baseplate position (12/12 cases of notching in case of high baseplate vs. 1/36 in case of low position; *p* < 0.001). Patients with and without notching did not show a significant difference in baseplate inclination (14° ± 9° vs. 16° ± 8°, *p* = 0.408). Glenoid RLL were significantly more frequent in patients with scapular notching (31% vs. 6% *p* = 0.038). Notching was not associated with GT (*p* = 0.611) and calcar resorption (*p* = 0.716). Patients with scapular notching presented lower clinical results (Table 3).

## 4. Discussion

This study showed that IB-RSA is a safe and effective procedure and does not present specific implant-associated complications at mid-term follow-up.

Scapular notching remain the most common complication associated with RSA [1,3]. In this series, notching occurred in 27% cases. Even though this rate still is not completely satisfying, it is lower compared to the notching rates (40–68%) of similar standard bearing RSA with a classic Grammont humeral stem with or without a lateralized glenoid [1,8,15,16,25]. Moreover, it must be underlined that the notching observed in this series has peculiar features. First, at this follow-up, no grade higher than 2 was observed (Figure 5).

As postulated by Friedman et al., grade 3 and 4 extending over the inferior screw are likely the results of a biologic response to polyethylene particles and osteolysis [3]. Secondly, in all cases with notching, a bone spur on the scapular neck was present. Third, the notching was almost only present in cases with a high position of the baseplate, a condition that is proven to be associated with mechanical contact of the prosthesis with the scapula [5,8,11,26]. All these features seem to be linked to a pure mechanical notching, proving that IB-RSA with a hard humeral liner leads to a distinct type of scapular notching and avoids PE wear-induced osteolysis at mid-term follow-up. Similar findings were shown by Kohut et al. using a different IB-RSA [27]. Based on our findings, optimal (as low as possible) and secure (optimal preparation of the subchondral bone) positioning of the glenoid is mandatory to avoid scapular notching (Figure 6).

Further studies are needed in order to analyze the notching evolution with IB-RSA at longer follow-up and verify if the notching remains mainly mechanical, or if the osteolysis will spread over the screw reaching the central peg, with a potential risk of loosening [12]. Moreover, histological studies on retrieved implants will be useful to clarify this phenomenon in vivo. In contrast with other authors [8,16], we did not find a statistical difference in baseplate inclination between patients with and without notching in our series. This finding may have different explanations. First, the limited number of patients included in this series compared to other series [8,16]. Second, the design of the glenosphere presenting an inferior extension may compensate for a slight superior inclination of the baseplate (Figure 2). Third, we believe that scapular notching is more linked to the inclination of the baseplate relative to the scapular neck [11] or to the intrinsic neck morphology [3] compared to the inclination relative to the supraspintus fossa.

Another interesting observation was that in this series, GT and calcar resorption were not associated with scapular notching, which contrasts the finding of Mazaleyrat et al. [28] using a standard bearing RSA. This observation can be related to the fact that with IB-RSA PE, wear-induced osteolysis is minimized with a potential effect on proximal humerus resorption. The high rate of humeral stress shielding observed in this series is likely due to the methaphyseal fixation of the stem, which is associated with these radiographical changes [24,28]. Regarding the higher rate of glenoid RLL among patients with notching, we believe that this finding is due to the presence in this group of the three patients with early subsidence with the development of non-progressive RLL and notching. However, further studies at longer follow-ups are necessary to clarify the evolution of this observation.

The impact of scapular notching on clinical function is controversial. Some authors did not find an influence of notching on clinical function [2,7,8,12], while other authors clearly stated that scapular notching is associated with lower functional scores and decreased range of motion [6,9,10,29], especially in case of high grades. In this series, notching development has a significant negative influence on functional scores and anterior elevation at mid-term follow-up. We believe that this finding is due to the fact that the notching observed with this prosthesis is mainly mechanical and strictly linked to the incorrect high position of the glenoid. This leads to premature contact of the humeral component with the scapula and the acromion and consequent limitation of the motion [26].

Clinical scores and range of motion were overall improved at mid-term follow-up and were comparable with other series of IB-RSA [27,30] or standard RSA [6,7,8,9,10,16]. Interestingly, patients with a 44 mm diameter showed significantly better external and internal rotation, better CS, and a positive trend for better anterior elevation. Biomechanical studies showed that increasing the glenosphere diameter may have a favorable effect on range of motion [31,32,33]. Clinically, some authors found similar improvements when increasing the diameter to 42 or 44 mm [34,35]. The use of bigger glenospheres is advisable in order to improve the results. However, this finding should be taken with caution, since it is not possible to use a 44 mm glenosphere in all patients due to technical issues.

Regarding complications, only one patient was revised because of an infection and two patients had an acromial fracture. The reported rate of acromial fracture with classic reverse design is generally lower [1,36], but due to the low number of patients included in this series, it is impossible to evaluate whether increasing the glenosphere diameter could be associated with a higher risk of acromial fracture. However, Kohut et al. [27], in a larger series comparing 40/44 mm with 36 mm glenosphere, did not report an increased risk of acromial fracture. One important phenomenon that we observed in three cases is an initial glenoid migration, which stabilized within the first year and remains in a high and severe superiorly tilted baseplate (Figure 2). We noticed that in all these cases, the superior part of the glenoid was not perfectly in contact with the subchondral bone, probably due to an incorrect technique and uncompleted cartilage removal. Based on this observation, we recommended a good preparation of the subchondral bone in order to match the baseplate profile.

This study presents the following strengths: the series included all consecutive patients prospectively enrolled during the index period; the same prosthesis and the same technique was used in all the cases; radiographical and clinical evaluation was performed by two surgeons (R.R. and L.D.) not involved in the surgical procedure. However, this study presents some limitations. Firstly, its retrospective nature. Secondly, the lack of a control group. To definitively confirm our observation, comparative studies comparing IB-RSA and standard RSA will be needed in the future. Thirdly, because of the limited number of patients included it is impossible to draw definitive conclusions regarding complications and revisions rate. Fourth, other anatomical measures (i.e., scapular neck angle, neck-shaft angle, glenoid inclination) that may have a role in scapular notching were not measured.

## 5. Conclusions

IB-RSA is a safe and effective procedure without specific implant-associated complications at mid-term follow-up. Overall, using a 44-mm-diameter glenosphere compared to 40-mm-lead to an improved range of motion. Inverting biomaterials lead to a distinct kind of notching, which mainly showed mechanical features and no observed cases of grade 3 or 4. Scapular notching is associated with a high baseplate position and has a negative influence on range of motion and clinical outcome.

## Figures and Tables

**Figure 1 jcm-11-05796-f001:**
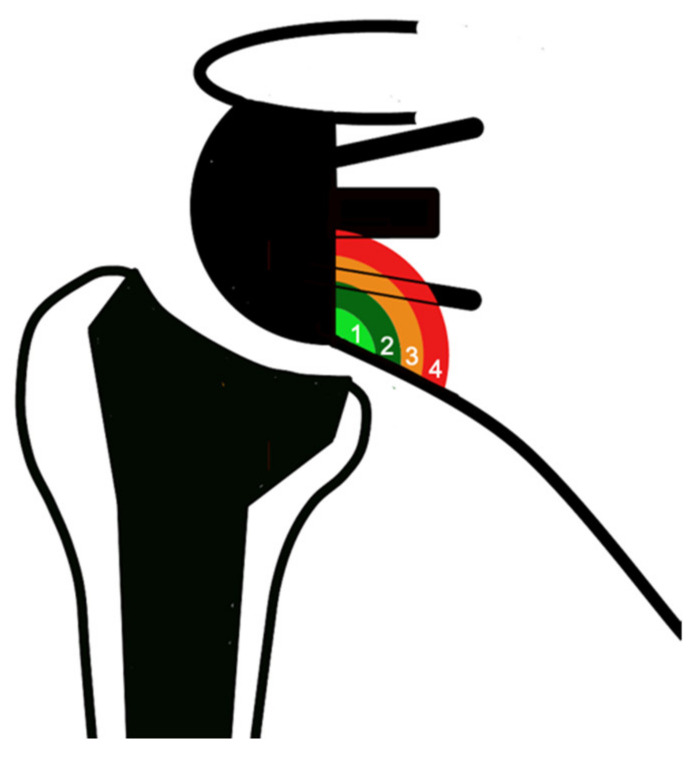
Notching classification according to Sirveaux et al. [6]: grade 1—defect confined to the pillar; grade 2—defect reaching the lower screw; grade 3—defect over the lower screw; grade 4—defect extended under the baseplate.

**Figure 2 jcm-11-05796-f002:**
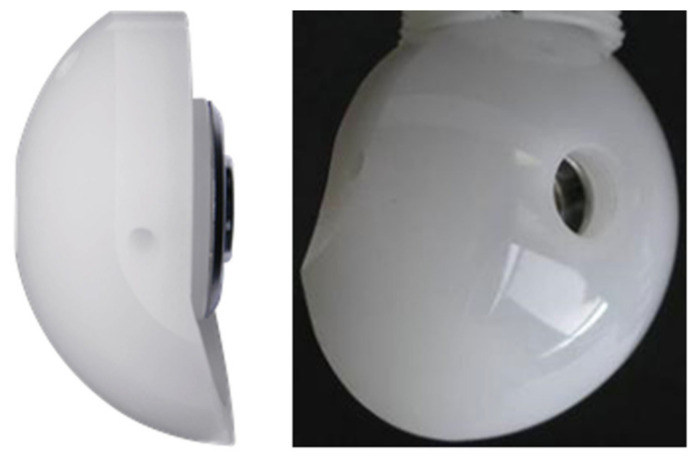
SMR reverse HP glenosphere (LimaCorporate S.p.A, 33038 Villanova di San Daniele del Friuli, Udine, Italy).

**Figure 3 jcm-11-05796-f003:**
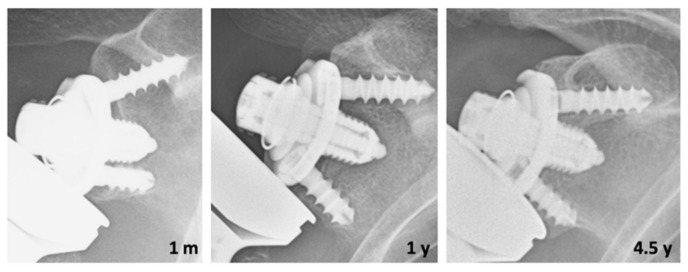
Subsidence of the glenoid due to incomplete glenoid preparation (baseplate not completely in contact with subchondral bone) which stabilizes at last follow-up in high glenoid position with the development of grade 2 scapular notching. m, months; y, year.

**Figure 4 jcm-11-05796-f004:**
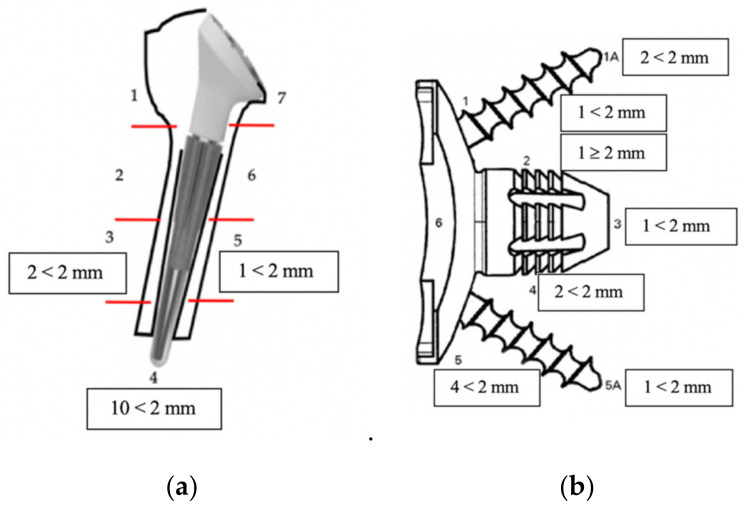
Image representing the frequency (number of cases) of RLL per zone of the humerus (**a**) and the glenoid (**b**).

**Figure 5 jcm-11-05796-f005:**
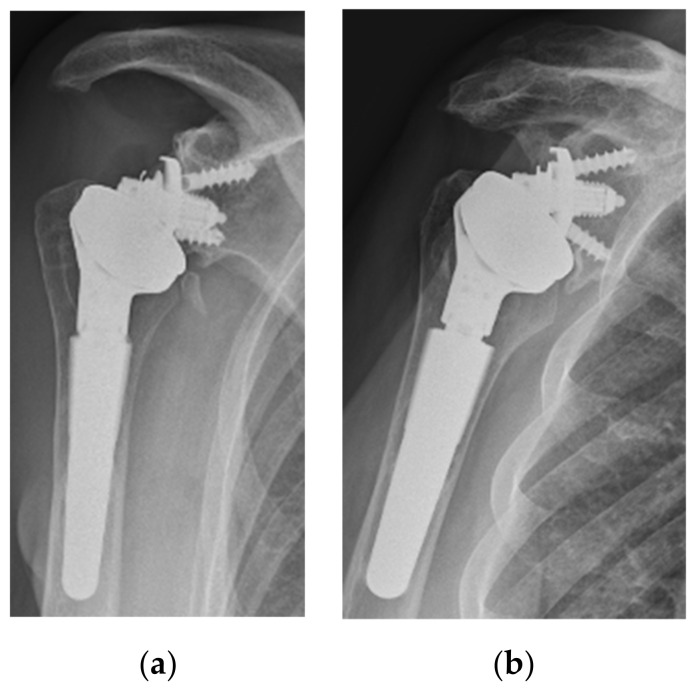
Two cases with a high glenoid position that developed a grade 1 (**a**) and grade 2 (**b**) of notching with the formation of a bone spur.

**Figure 6 jcm-11-05796-f006:**
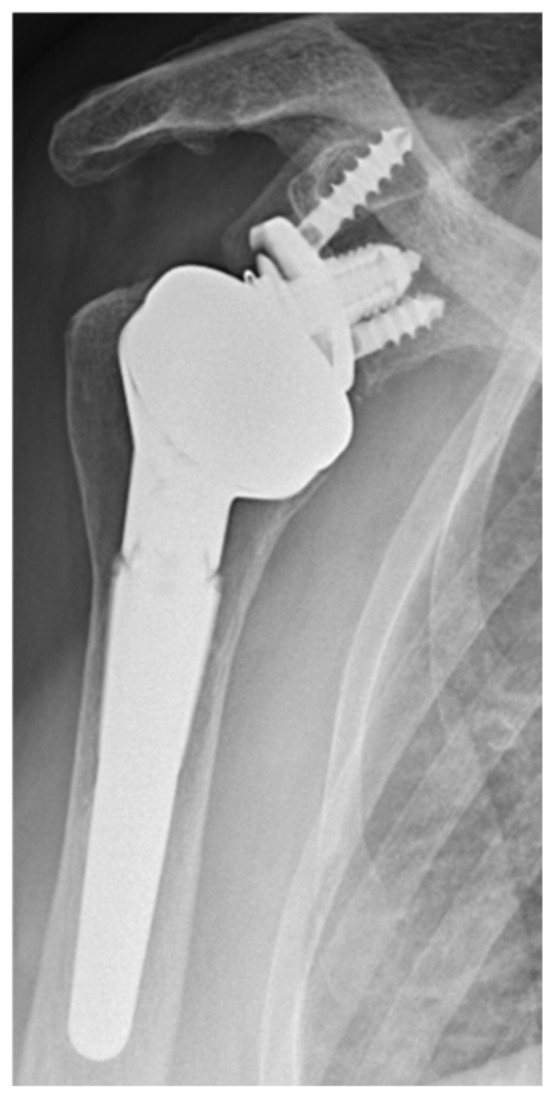
Correct position of the glenoid.

**Table 1 jcm-11-05796-t001:** Preoperative and postoperative clinical outcomes.

Outcome	Preop	Postop	*p* Value
CS	23 ± 13	67 ± 17	<0.001
ASES	37 ± 21	81 ± 18	<0.001
SSV	27 ± 24	77 ± 16	<0.001
Pain	7.3 ± 2.4	1.0 ± 1.8	<0.001
AE	66° ± 37°	140° ± 32°	<0.001
ER	15° ± 14°	26° ± 17°	0.042
IR	3.9 ± 2.1	5.4 ± 2.9	<0.001

CS, Constant Score; ASES, American Shoulder and Elbow Surgeon; SSV, Subjective shoulder value; AE; anterior elevation; ER, external rotation; IR, internal rotation.

**Table 2 jcm-11-05796-t002:** Clinical outcomes according to glenosphere size. No preoperative or demographical differences were found between the two groups.

Outcome	40 mm (38)	44 mm (10)	*p* Value
CS	63 ± 17	78 ± 11	**0.006**
ASES	79 ± 19	87 ± 15	0.206
SSV	75 ± 17	84 ± 12	0.141
AE	133 ± 33°	157 ± 19°	0.051
ER	23 ± 17°	36 ± 17°	**0.036**
IR	4.8 ± 2.7	7.8 ± 2.2	**0.011**

CS, Constant Score; ASES, American Shoulder and Elbow Surgeon; SSV, Subjective shoulder value; AE; anterior elevation; ER, external rotation; IR, internal rotation.

**Table 3 jcm-11-05796-t003:** Clinical outcomes according to notching at the last follow-up. No preoperative or demographical differences were found between the two groups.

Outcome	No Notching (35)	Notching (13)	*p* Value
CS	70 ± 15	58 ± 20	**0.046**
ASES	86 ± 14	70 ± 22	**0.031**
SSV	81 ± 14	68 ± 20	**0.027**
Pain	0.9 ± 1.9	1.2 ± 1.6	0.058
AE	148° ± 23°	115° ± 37°	**0.006**
ER	28° ± 15°	20° ± 21°	0.142
IR	5.7 ± 2.5	5.1 ± 3.6	0.561

CS, Constant Score; ASES, American Shoulder and Elbow Surgeon; SSV, Subjective shoulder value; AE; anterior elevation; ER, external rotation; IR, internal rotation.

## Data Availability

Not applicable.

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
