# Peer review of "Inverted-Bearing Reverse Shoulder Arthroplasty: Consequences on Scapular Notching and Clinical Results at Mid-Term Follow-Up"

_jcm, 2022, doi:10.3390/jcm11195796_

Round 1

Reviewer 1 Report

Very good paper. It touches upon the problem of reverse shoulder prosthesis, which is more and more often used in the clinical world. Moreover, one of the most common problems - scapular notching. All sections of the paper are written professionally, logically and to the point. However, it should be remembered that the article will be published in a journal that is read not only by shoulder surgeons. Therefore, in the introduction, I propose to include a pictorial photo / figure illustrating the notching phenomenon in a simple way

Reviewer 2 Report

General comments

The authors aim to report the results of IB-RSA on a consecutive series of patients at mid-term follow-up focusing on the incidence of grade 3 and 4 scapular notching.

This is an interesting study, and the authors must be congratulated for the work that they have carried out.

One could regret the lack of in-depth analysis especially regarding the position of the glenoid. The authors turn their article around the question of the notching and the mechanic/biologic theory. However, more analysis must be done to answer this question and moreover to establish a causality relation between the position of the implant (especially inclination), the inverted design with PE on the glenoid and the notching rate. The article loses its interest without it.

Multiple form issues have also to be corrected.

Specific comments

Title : it should evocate the notching and it seems too generic.

Abstract:

As it is a retrospective study, the analyzed population (49 patients) must be given in the methods and non-included patients among the 78 must be justified.

Complications and revisions must be presented first. We suppose that the revised shoulder has not been included in the functional analysis?

“A 44mm glenosphere was associate with better …”: better than what? Please be more specific.

Introduction:

Well written

Methods:

Line 82: “it is characterized by an inversion of the materials to reduce polyethylene debris, and, by a smart design to facilitate the implantation and improving range of motion”: these are subjective arguments. Please be more nuanced regarding “to reduce polyethylene debris” (that has not been proven) and “smart design” (that is a marketing argument).

Line 87: “The glenosphere presents also a 4mm eccentricity option, but at our institution it’s not utilized in primary cases.”: if it is not utilized in the present study, it has to be removed.

Line 88: “Humeral stem is usually implanted with 0° of retroversion.”: usually? What does it mean? This is an important point to address as it could have an impact on the notching. Was 0° the target? Do the surgeons use the ancillary or have another technic of implantation?

Line 103: on which XRays was this evaluated ? (postop, last follow up?)

Line 124: “c2”? Maybe Chi2

Results:

Line 127: “Among the 78 patients, 6 were dead, 17 were lost or impossible to contact, 6 refused the control. Forty-nine patients were reviewed clinically and radiologically at a mean follow-up of 5.0 ± 0.9 years.” And the description of the population must figure in the Methods section. A table would be welcome in the Methods section too.

Please make the reader understand 1) the patients included and not included (in the Methods, after removing the lost to follow up, dead patients or disagreed patients, 2) the patient excluded of the functional analysis (in the Results section) for revision.

Indications, epidemiologic data, implants,.. must figure in the Methods too in the Table.

Complications and Revisions must be the first section of the Results, then the clinical and radiological evaluation.

Line 137: “At the last follow-up CS improved 23 ± 13 from to 67 ± 17 (P<0.001), ASES from to 137 (P<0.001), SSV from 27 ± 24 to 77 ± 16 (P<0.001), pain from 7.3 ± 2.4 to 1.0 ± 1.8 (P<0.001). AE increased from a mean of 66° ± 37° to 140° ± 32° (P<0.001), ER1 from 15° ± 14° to 26° ± 139 17° (P=0.042), IR from 2.3 ± 2.6 to 5.4 ± 2.9 (P<0.001).” please, make a Table.

Number of significant digits must be respected (no number after the comma).

How was the size of the glenosphere chosen? Please specify in the Methods section. It could be an important biase.

Line 148: Acromial fractures must appear in the Complications/Revisions section.

Same for the infection, apraxia...

Line 160: “It hesitated in high glenoid 160 position with development of grade 2 scapular notching” What does it mean? Please specify.

A specific section for notching should be done.

Line 173: « Glenoid RLL were significantly more frequent in patients with scapular notching. »: this goes against the “mechanic” theory as the glenoid lucency is related to PE debris. Please justify in the discussion.

Figure 4: it seems that more than the problem of glenoid upper/lower position, the inclination of the glenoid implant may play a role. Indeed, this figure shows an upward inclinated baseplate (regarding to the sclerotic line of the supraspinatus fossa). Did the authors analyzed this point?

Line 197: same comment.

Line 213: what about the important RLL rate observed in some patients?

Line 227: this is the point. Inclination (and RSA angle) has to be analyzed. If not, the article will be highly criticable regarding its hypothesis on the mechanic/biologic theory and the real interest of inverting the PE and metal implants.

Line 234: indeed, however a direct analysis between the size of the glenosphere and the notch must be carried out, ideally by adjusting patient on the sex.

A conclusion with a take home message would be welcome.

Round 2

Reviewer 2 Report

The article has been correctly improved and is interesting. 

Congratulations to the authors.